# Widespread misidentification of scanning electron microscope instruments in the peer-reviewed materials science and engineering literature

**Reese A. K. Richardson**[1,2], **Jeonghyun Moon**[1,2], **Spencer S. Hong**[1,3], **Luís A. Nunes Amaral**[1,2,4,5,6,7]*

**1** Department of Engineering Sciences and Applied Mathematics, Northwestern University, Evanston, Illinois, United States of America, **2** Department of Molecular Biosciences, Northwestern University, Evanston, Illinois, United States of America, **3** Department of Chemical and Biological Engineering, Northwestern University, Evanston, Illinois, United States of America, **4** Division of Pulmonary and Critical Care, Northwestern University Feinberg School of Medicine, Chicago, Illinois, United States of America, **5** Department of Physics and Astronomy, Northwestern University, Evanston, Illinois, United States of America, **6** Northwestern Institute on Complex Systems (NICO), Northwestern University, Evanston, Illinois, United States of America, **7** NSF-Simons National Institute for Theory and Mathematics in Biology (NITMB), Chicago, Illinois, United States of America

* amaral@northwestern.edu

**Data availability statement:** All data presented in this work which can be published according

## Abstract

Materials science and engineering (MSE) research has, for the most part, escaped the doubts raised about the reliability of the scientific literature by recent large-scale replication studies in psychology and cancer biology. However, users on post-publication peer review sites have recently identified dozens of articles where the make and model of the scanning electron microscope (SEM) listed in the text of the paper does not match the instrument's metadata visible in the images in the published article. In order to systematically investigate this potential risk to the MSE literature, we develop a semi-automated approach to scan published figures for this metadata and check it against the SEM instrument identified in the text. Starting from an exhaustive set of 1,067,108 articles published since 2010 in 50 journals with impact factors ranging from 2 to 24, we identify 11,314 articles for which SEM manufacturer and model can be identified in an image's metadata. For 21.2% of those articles, the image metadata does not match the SEM manufacturer or model listed in the text and, for another 24.7%, at least some of the instruments used in the study are not reported. We find that articles with SEM misidentification are more likely to have existing observations of improprieties made on post-publication peer review site PubPeer than other MSE articles and that a subset of these articles within the subfield of electrochemistry are more likely to incorrectly estimate the optical band gap if the article features SEM misidentification. This suggests that SEM misidentification may be a tractable signature for flagging problematic MSE articles. Unexplained patterns common to many of these articles suggest the involvement of paper mills, organizations that mass-produce, sell authorship on, and publish fraudulent scientific manuscripts at scale.

to the copyright information and text and data mining licenses of each publisher surveyed and which is not under license from the PubPeer Foundation is available in Supplementary Materials. Data from OpenAlex is publicly available. Identifiers and labels of each article analyzed are available at https://doi.org/10.17605/OSF.IO/6C5RF.

**Funding:** R.A.K.R. was supported in part by the National Institutes of Health Training Grant [T32GM008449] through Northwestern University's Biotechnology Training Program. R.A.K.R. gratefully acknowledges funding from the Dr. John N. Nicholson fellowship from Northwestern University; Moderna Inc., Identifying bias and improving reproducibility in RNA-seq computational pipelines. J.M. gratefully acknowledges funding from the Weinberg College Baker Program in Undergraduate Research. S.S.H. gratefully acknowledges support from the Ryan Fellowship and the International Institute for Nanotechnology at Northwestern University. L.A.N.A. gratefully acknowledge funding from SCISIPBIO: a data-science approach to evaluating the likelihood of fraud and error in published studies [1956338].

**Competing interests:** The authors declare that they have no competing interests.

## Introduction

Trust in published peer-reviewed results is critical to the efficient operation of the scientific enterprise. It is implicitly assumed in the peer-review process that the authors of a study have, to the best of their abilities, carefully and systematically reviewed their work for potential flaws and that no deception is being attempted. Such assumptions are sometimes spectacularly violated, as in the infamous case of J. H. Schön, but are nonetheless thought to hold broadly. It is thus not surprising to see the widespread concern raised by recent large-scale replication studies [1–3] that have found many published results in the social sciences and biology to be irreproducible. Despite Schön's case and other recent high-profile retractions in superconductivity research [4], lack of replicability of results in the physico-chemical sciences and engineering has not been broadly regarded as concerning. However, two studies have recently reported widespread irreproducibility in the characterization of metal-organic frameworks [5] and hydrogen storage materials [6].

Amid a broader "reproducibility crisis" [7] and an exponentially-growing scientific literature [8], there has been an increasing interest in developing approaches for detection of markers of irreproducibility, errors and impropriety at-scale. For instance, Labbé and colleagues developed *Seek & Blastn* to rapidly screen publications for misidentified nucleotide reagents [9–11]. Bik and colleagues manually screened over 20,000 published articles in the life sciences for image duplication [12], finding that 3.8% of published articles contained inappropriately duplicated figure elements. Semi-automated approaches that accelerate the discovery of image integrity issues have also been developed [13–15]. Several publishers now employ automated methods for detecting image duplication in unpublished manuscripts [16].

In an effort to extend the development of automated or semi-automated tools in the context of the physico-chemical sciences and engineering, we study here the reporting of scanning electron microscope (SEM) instrumentation in scientific papers in material science and engineering, broadly defined. SEMs are a critical tool for the characterization of samples [17,18]. Tens of thousands of articles using SEM are published annually (Fig 1a and S1 Fig). Studies reporting on original research articles will identify the manufacturer, model and operating parameters of the SEM used — e.g. "samples were observed with a Philips XL30 field emission scanning electron microscope at an accelerating voltage of 10 kV." Many times, the published images obtained with the SEM will include the instrument's auto-generated "banner" that discloses experimental metadata (Fig 1b). Depending on the SEM manufacturer, this banner can include information on the instrument's manufacturer and model (Fig 1c), various operating parameters of the instrument, and the facility that operates the instrument.

Recently, several users on the post-publication peer review site PubPeer [20] have documented dozens of instances of articles where the SEM instrument listed within the manuscript's text does not match the instrument's metadata extractable from the published images [21]. Such extreme carelessness suggests that other things may be amiss with the study. It is difficult to imagine such a mistake being made in good-faith preparation of an article. However, if articles are hastily mass-produced, or the article text was plagiarized from one source and the article images from another, these would be among the likely details to be missed. Indeed, PubPeer commenters often note other inconsistencies in these articles that directly call into question the reliability of the findings or the provenance of the article.

To systematically investigate this matter, we developed a semi-automated pipeline for detection of misreporting of SEM instrumentation in published research articles. We deployed our method on more than a million articles from across 50 journals published by 4 different publishers, nearly 175 thousand of which included SEM images (Fig 1d). While we

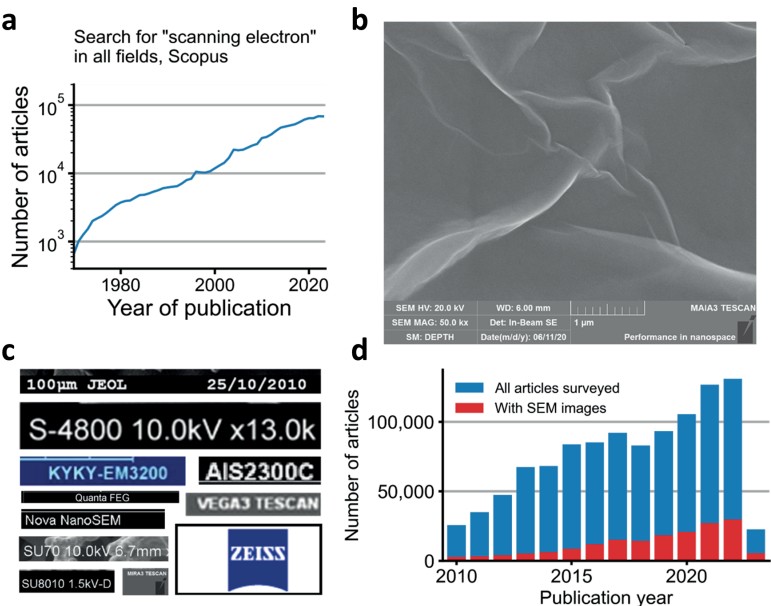

**Fig 1. We developed a semi-automated pipeline for detection of misidentified SEM instruments in the MSE literature and tested it on published articles in 50 journals**. **a,** Annual count of articles featuring SEM based on Scopus search results. **b,** SEM image featuring an auto-generated metadata banner from a TESCAN MAIA3 SEM, after [19]. **c,** Illustrative metadata banner snippets allowing for identification of the manufacturer and/or model of instrument used. **d,** Publication timeline of the 1,067,108 articles extracted from 50 journals between 2010 and early 2023. SEM image were present in 174,046 articles (16.3%).

were only able to extract SEM metadata for approximately 11 thousand articles, we found that for 21.2% of these articles the image metadata did not match the SEM manufacturer or model listed in the text of the manuscript. For another 24.7%, at least some of the SEM instruments used in the study were not disclosed.

## Data and methods

In order to produce a precise estimate of the rate at which misidentification of SEM instrumentation occurs in materials science and engineering literature overall, one would need to obtain a representative sample of journals appropriately stratified by publisher, impact, reputation, subfield, author characteristics, and so on. Unfortunately, several factors precluded achieving this goal. First, several large publishers of high-profile materials science journals have established 'data mining' licensing terms and conditions that limit at-scale downloading to only open-access content and to PDF files. Second, many publishers severely limit request rates, precluding comprehensive downloading of the articles they publish. Third, the sensitivity of automated approaches is dependent on the image size of published scientific images. Unless images are sufficiently large and high-resolution, metadata banners cannot be automatically identified. The size of images published as figures has changed substantially over time and varies considerably across publishers (Fig 2 and S2 Fig). Finally, SEM instrument misidentification is only detectable where metadata banners are visible in the published figures. In the vast majority of materials science articles where metadata banners are not visible, rates of instrument misidentification are entirely unknowable.

Due to these constraints, we focused on fifty journals from four publishers that (i) are indexed in OpenAlex, a free and open-source index of the scientific literature [22], (ii) span

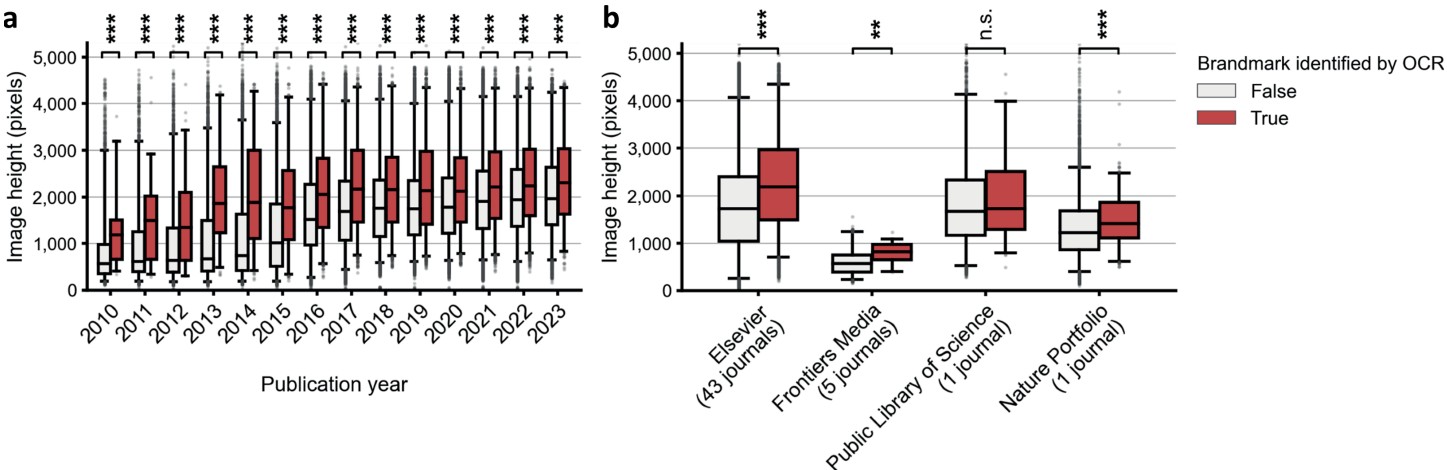

**Fig 2. Comparison of the sizes of images, in terms of image file height in pixels, from which we were able to extract brandmarks using optical character recognition (OCR) (red boxes) versus images for which we were not (white boxes). a,** For all years, image files for which we were able to extract brandmarks were larger. **b,** Comparison across 4 publishers of the sizes of images for which we were able to extract brandmarks (red boxes) versus images for which we were not (white boxes). Except studies published by the Public Library of Science, images for which we were able to extract brandmarks were significantly larger. Center line shows median, boxes show inter-quartile range, whiskers show 2.5th percentile and 97.5th percentile. n.s. = $p > 0.05$, * = $p < 0.05$, ** = $p < 0.01$ and *** = $p < 0.001$ by two-sided Mann-Whitney U test. Additional analysis shown in S2 Fig.

a broad range of impact factors, (iii) span a broad range of subfields within materials science and engineering and (iv) for which there is evidence of research impropriety based on existing comments on PubPeer. All content was downloaded following the relevant data mining terms, conditions and licenses specified by the publishers.

We focus heavily on Elsevier journals due to the publisher's relatively permissive data mining terms and conditions. We also included journals published by three additional publishers: Frontiers Media, Springer Nature, and the Public Library of Science. Several journals were multidisciplinary in scope, while others were focused on chemistry, materials science, chemical engineering, optics, biomedicine, and construction engineering. All but two journals were indexed by Web of Science [23] with 2022 Journal Impact Factors (IFs) ranging between 2.1 and 24.2. Journal characteristics are listed in S1 Table.

We crawled the webpage of all 1,067,108 articles published in the 50 journals since 2010 using the digital object identifiers (DOIs) obtained from OpenAlex (downloaded Feburary 27, 2023). For each article, we scanned the figure captions for terms indicative of SEM usage such as "sem micrograph" and "scanning electron micro" (see S2 Table for a complete list). This filtering step identified 174,046 articles with an average of 1.5 SEM-related figures per article. We downloaded the largest available versions of each of these figures. We next used Tesseract 4 [24] optical character recognition (OCR) at 1.0, 1.5 and 2.0 scale to process each of the obtained 267,603 figures. We then compared the extracted strings against a manually curated set of brandmarks and model numbers known to appear in SEM metadata banners (Fig 1c). Examples include "TESCAN", "KYKY-EM3200", "ZEISS", and "SU8010" (see S3 Table for a complete list). For 11,314 of 174,046 articles articles with SEM related images, we were able to match the OCR output to one of the brandmarks in our set. We next downloaded the full text of the 11,314 articles identified in the previous step. For each article's full text, we extracted portions of the text around occurrences of SEM brands (see S4 Table for a complete list of patterns).

After completing this automated pipeline (Fig 3), R.A.K.R and J.M. manually compared the brandmark and model extracted from the OCR of the images to the brandmark and model listed in the full text of the article identified in the text snippets. We chose to perform manual labelling because we found it to be a more straightforward approach than implementing an automated pipeline. We found that, after some practice, we could execute this task at a rate of 5–10 seconds per article.

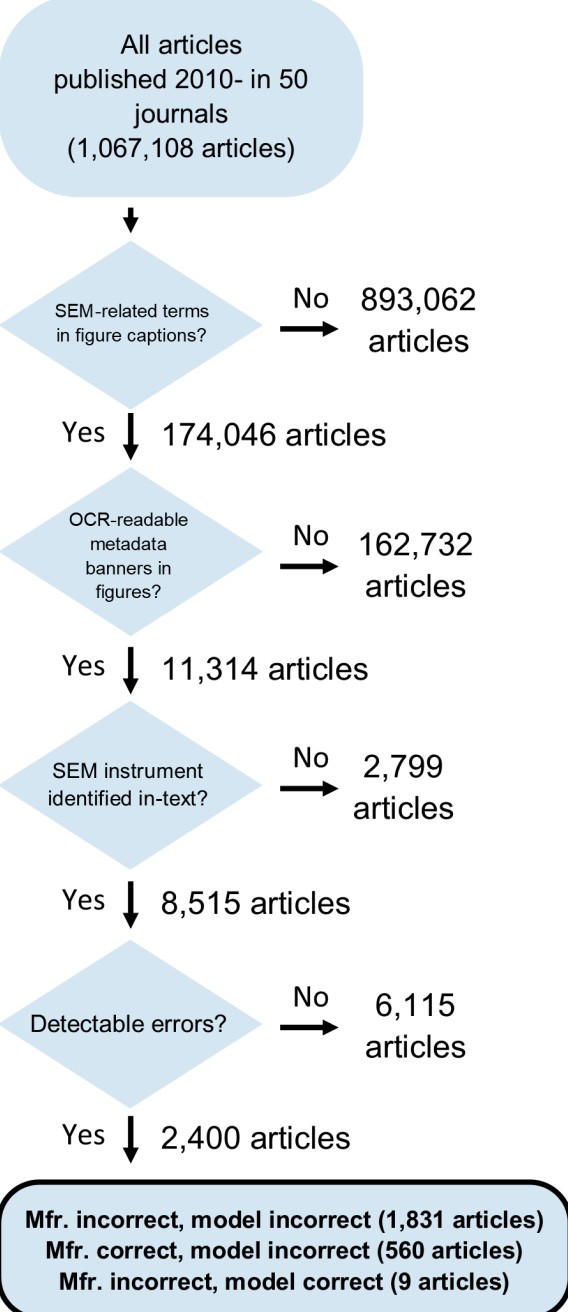

**Fig 3. Flowchart of implemented semi-automated pipeline.**

We determined for each article whether the SEM used was declared in the full text. If this was the case, we also determined whether the brandmark of the SEM declared in the full text correctly matched the metadata of the instrument extracted from the images. If this was the case, we also determined whether the model of the SEM declared in the full text correctly matched the model of the instrument extracted from the images.

In order to check for false positives introduced by our pipeline and to determine whether there could be a good explanation for the misidentification of SEMs in the detected articles, we selected a random sample of 150 articles from the set of detected articles labeled as problematic by our pipeline. S.S.H. reviewed each of the 150 articles (full text, figures and supplementary material) for mislabeling on our part and possible justifications for what would appear to be instrument misidentification.

To assess whether articles with SEM misidentification were more likely to feature other errors, we looked for a specific common error in the estimation of the optical band gap from Tauc plots (Fig 6). We searched the full text of all 11,314 articles with identifiable SEM brandmarks for the string 'tauc'. Our search returned 644 articles. 345 of 356 articles for which we had detected no errors in SEM identification included Tauc plots; 123 of 128 articles for which SEM was not declared in text included Tauc plots; and 154 of 160 articles for which the SEM instrument was incorrectly declared in the text included Tauc plots. To complete our analysis, we randomly sampled an additional 300 articles out of the 167,732 with SEM images but no identifiable SEM bandmarks, 283 of which included Tauc plots. We did not assess articles without SEM images since we did not download their full texts.

We manually screened the main text and supplementary materials of every sampled article, annotating whether they had Tauc plots and flagging them if there was at least one Tauc plot where the band gap was improperly estimated because of the error described in Fig 6). Note that we did not assess for other errors (such as if the linear fit was not applied to the linear portion of the curve). We reproduce our annotations for the collected 944 articles in S6 Table.

In order to estimate the scientific impact of the identified articles with SEM misidentification, we downloaded number of citations to each article in Scopus [25] using *pybliometrics* v3.5.2 [26] on January 29, 2024. We found Scopus records for 11,304 of 11,314 articles with SEM brandmarks and determined the presence of PubPeer comments on each article using a Feb 1, 2024 snapshot of PubPeer.

Finally, in order to estimate characteristics of the authors of the identified articles with SEM misidentification, we downloaded author names and affiliations as reported on the article page and obtained country of of authorship from the affiliation strings. We did not, however, attempt automated author disambiguation.

Code is available at *github.com/amarallab/sem_images*.

## Results

We employed a semi-automated pipeline to identify articles with misidentified SEM instruments (Fig 3). As part of this pipeline, we applied OCR to figures containing SEM images from 174,046 peer-reviewed articles (Fig 1d). We noticed that our OCR engine sometimes missed images in which brandmarks were identifiable by eye. The accuracy of OCR is known to be resolution-dependent [27] and image sizes varied considerably with time (Fig 2a and S2 Fig) and across publishers (Fig 2b and S2 Fig). This suggests that our approach may underestimate the global rate of recognizable SEM instrument misidentification among articles containing SEM images, especially for journals with low standards for sizes of images published as figures.

We identified 11,314 articles with SEM images containing OCR-readable brandmarks. Of these, 6,115 articles identified a SEM in the text that match the one identifiable from a figure (54.0%), 2,799 did not identify in the text the SEM used (24.7%), and 2,400 articles identified a SEM in the text that did not match the one identifiable from a figure (21.2%, Fig 4a). Of those in the latter group, 1,831 articles had mismatches for both manufacturer and model, and 569 articles had mismatches for either the manufacturer or the model.

Fig 4b shows the number of articles with SEM misidentification for the fifteen journals with the highest numbers and S1 Table presents the full results by journal. We provide DOIs and our labels for all 11,314 articles in S5 Table.

## Methodological sensitivity and selectivity

If a single SEM instrument was used in an article, and we are able to extract manufacturer and model information from a figure and from the manuscript's text, then the problem of determining misidentification is straightforward. However, there could be cases where an article includes SEM images from multiple different instruments but only one of them is specified in the text. In order to check for the prevalence of such cases and to determine the error rate of our semi-automated pipeline, we selected a random sample of 150 articles identified as problematic by the pipeline and manually reviewed each article's reporting of SEM equipment. Out of these 150 articles, we found only 1 instance (0.67%; 95% CI = [0.16%, 2.4%]) where an error was introduced by our pipeline (during the manual labelling step) and the article did not in fact misidentify its SEM equipment. This analysis suggests that our semi-automated pipeline has a low false discovery rate.

We found an additional 32 cases (21.3%; 95% CI = [16% , 28%]) where images indicated use of multiple SEMs but in which the article text did not list all of the instruments identifiable in the images (Fig 5a). While this situation could plausibly explain the lack of agreement we initially identify, it still reveals an improper disclosure of all experimental conditions.

Of the 2,400 articles we identified as having misidentified instruments, only 43 (1.8%) had prior PubPeer comments (Fig 5b). Thirty (69.8%) had comments unrelated to SEM misidentification. Issues identified by PubPeer commenters include image duplications and

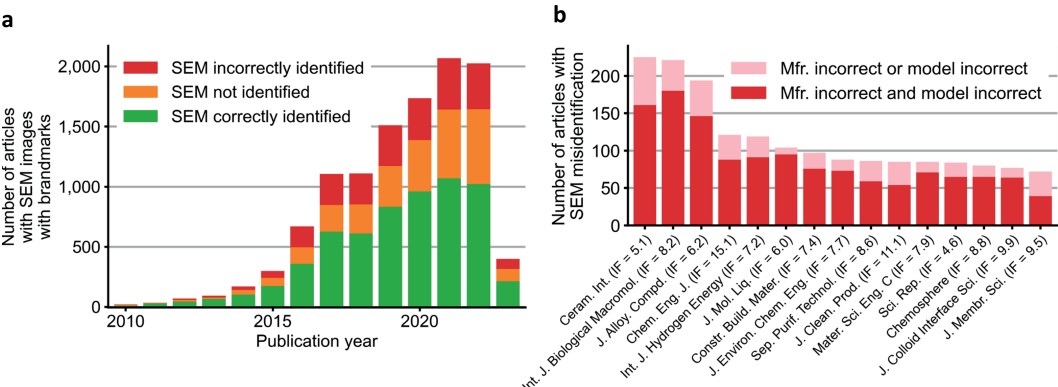

**Fig 4. Summary of instrument identification for 11,314 articles with SEM images with extractable metadata banners. a,** Publication timeline of the articles analyzed. 6,115 (54.0%) correctly identified the SEM used, 2,400 (21.2%) articles incorrectly identified the SEM used, and 2,799 (24.7%) articles did not identify the SEM used. **b,** Top 15 surveyed journals with the most articles with SEM misidentification. Most articles with SEM misidentification incorrectly identified both the manufacturer (Mfr.) and the model of SEM used.

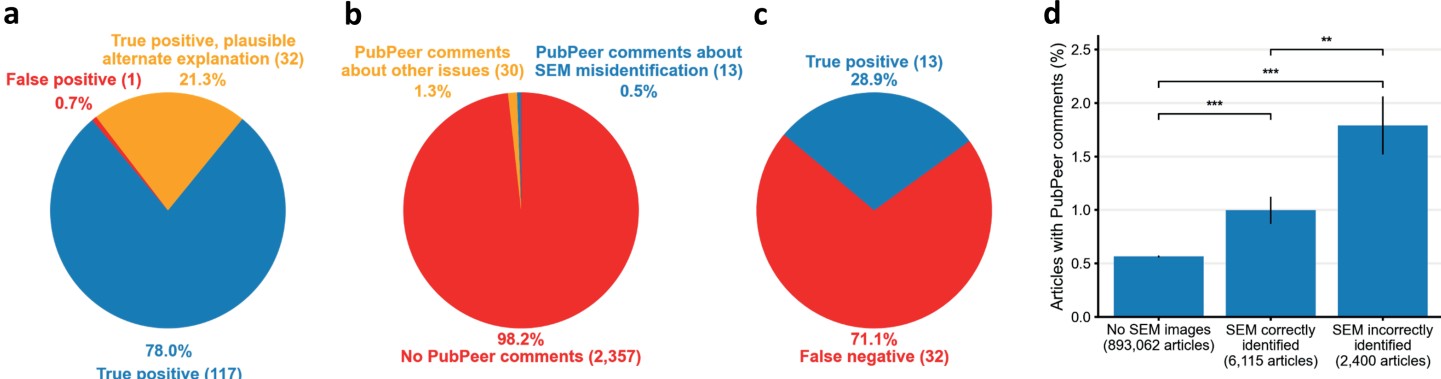

**Fig 5. Assessments of methodological sensitivity and selectivity. a,** Estimation of our pipeline's false discovery rate. We manually assessed a subset of 150 articles with SEM misidentification. 1 article (0.7%) was incorrectly classified during the manual step of our pipeline. 32 articles (21.3%) had fewer instruments identified in the text than were identified in the figures. **b,** Comparison of the results of our pipeline to PubPeer comments. Only 43 out of 2,400 articles labeled by our pipeline as problematic were already commented on by PubPeer users. For 30 of the 43, the issues commented upon were unrelated to SEM misidentification. **c,** Estimation of our pipeline's false negative rate. We determined the ability of our pipeline to recover all articles for which a PubPeer user had already reported SEM misidentification. Our pipeline caught 13 of 45 eligible articles (28.9%). **d,** Pre-existing Pubpeer comments were more frequently found on articles with misidentified SEM instruments than other MSE articles. Errors bars show ±1 standard error of the proportion. n.s. = p > 0.05, * = p < 0.05, ** = p < 0.01 and *** = p < 0.001 by two-sided Z-test of proportions.

manipulation of X-ray diffraction (XRD) and electron dispersive X-ray spectroscopy (EDX) spectra.

To assess the false negative rate of our pipeline, we collected 45 articles that we had already screened that also had prior PubPeer comments indicating SEM misidentification. Out of 45 instances of SEM misidentification noted on PubPeer, our pipeline caught thirteen (28.9%; 95% CI = [15.6%, 42.1%], Fig 5c). For 24 out of 32 studies, we failed to catch the issue because brandmarks were not extractable using Tesseract. For another 3 studies, Tesseract extracted the instrument manufacturer but not the instrument model. Finally, for 5 studies, we failed to recognize the presence of SEM images due to atypical language in figure captions. These results suggest that our pipeline has a high false negative rate. This suggests that SEM misidentification could be more than twice as frequent as what we detect with our pipeline.

Validating our hypothesis that improper identification of SEM instruments portends critical deficiencies, we found that prior PubPeer comments were about 1.8 times as frequent on articles that we found to have misidentified their instruments than on articles we found to have correctly identified their instruments (two-tailed Z-test of proportions, p <0.01) and 3.2 times as frequent on articles with SEM misidentification than on articles without SEM images (p <0.001, Fig 5d and S3 Fig).

To further validate, we sought to determine the frequency of a specific, critical methodological error in articles with and without SEM misidentification. This error concerns Tauc plots, a popular method in electrochemistry by which the optical band gap of a material can be estimated from absorption or diffuse reflectance spectra [28,29]. By this method, the energy of incident photons $h\nu$ is plotted on the x-axis and a transformation of the absorption coefficient $(\alpha h\nu)^{1/\gamma}$ is plotted on the y-axis. A linear fit is then made to the linear portion of the curve following the equation

$$(\alpha h\nu)^{1/\gamma} = A(h\nu - E_g) \tag{1}$$

where $A$ is a constant and $E_g$ represents the optical band gap energy of the material. By this equation, $h\nu = E_g$ if and only if $(\alpha h\nu)^{1/\gamma} = 0$. In other words, one can estimate the band gap

energy of the material by reading the x-intercept of this linear fit. However, it is a common mistake to calculate $E_g$ not from the intercept of the linear fit with the line $(\alpha h\nu)^{1/\gamma} = 0$, but instead from the intercept of the linear fit with the line $(\alpha h\nu)^{1/\gamma} = y_{\min}$, where $y_{\min}$ is the bottom limit of the y-axis as plotted (Fig 6a). An example of this error from a published article (since retracted) is provided in S4 Fig.

We manually inspected a sample of articles featuring SEM images and Tauc plots and found that this specific error was about 1.8 times as frequent on articles that we found to have misidentified their instruments than on articles we found to have correctly identified their instruments (two-tailed Z-test of proportions, p <0.01) and 2.8 times as frequent on articles with SEM misidentification than on articles with SEM images without identifiable brandmarks (p <0.001, Fig 6b, S6 Table).

### Systemic characteristics

Next, we explore potential systemic characteristics of the corpus of articles with misidentified SEMs. Collectively, these articles have been cited 83,781 times (mean 34.9 citations per article, median 22, minimum 0, and maximum 848). These values were slightly higher than the 32.4 citations per article (median 20, minimum 0, and maximum 1,768) observed for the set of articles with SEM images containing OCR-readable brandmarks with no SEM misidentification (two-sided Mann Whitney U test $p<0.001$).

To understand topical preferences of articles that misidentified their SEM instruments, we compared the words included in titles of articles with SEM misidentification to the words included in titles of articles containing SEM images but without SEM misidentification. Several terms were enriched in articles with SEM misidentification, including 'nanocomposite', 'green', 'nanoparticles', and 'chitosan' (Fig 7a).

We successfully extracted authors names and affiliations for 172,921 of 174,046 articles with figures containing SEM images, and 2,391 of 2,400 articles with misidentified SEMs.

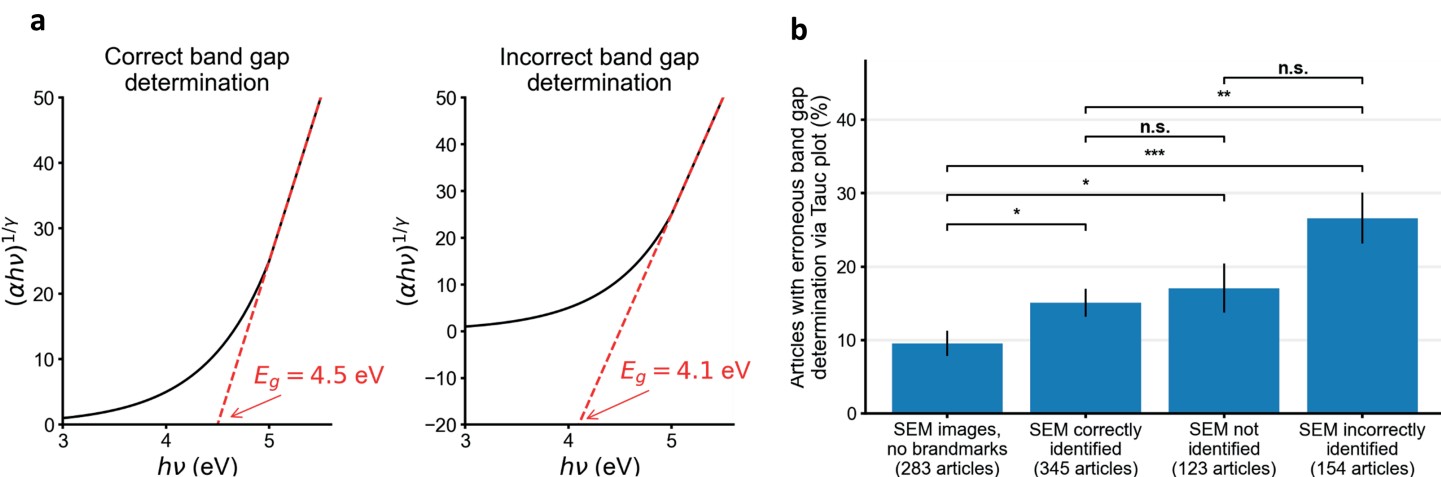

**Fig 6. Assessment of Tauc plots in articles flagged by our pipeline. a,** The mockup Tauc plot on the left demonstrates how to correctly estimate band gap energy (by taking the intersection of the plots linear fit with the $y \equiv (\alpha h\nu)^{1/\gamma} = 0$ line) [28]. The mockup Tauc plot on the right demonstrates a common error where the band gap is estimated by taking the intersection of the linear fit with the line $y = y_{\min}$ where $y_{\min}$ is the lowest value on the y-axis as plotted. **b,** This specific error occurred more frequently on articles with misidentified SEM instruments than other MSE articles. Errors bars show ±1 standard error of the proportion. n.s. = p > 0.05, * = p < 0.05, ** = p < 0.01 and *** = p < 0.001 by two-sided Z-test of proportions.

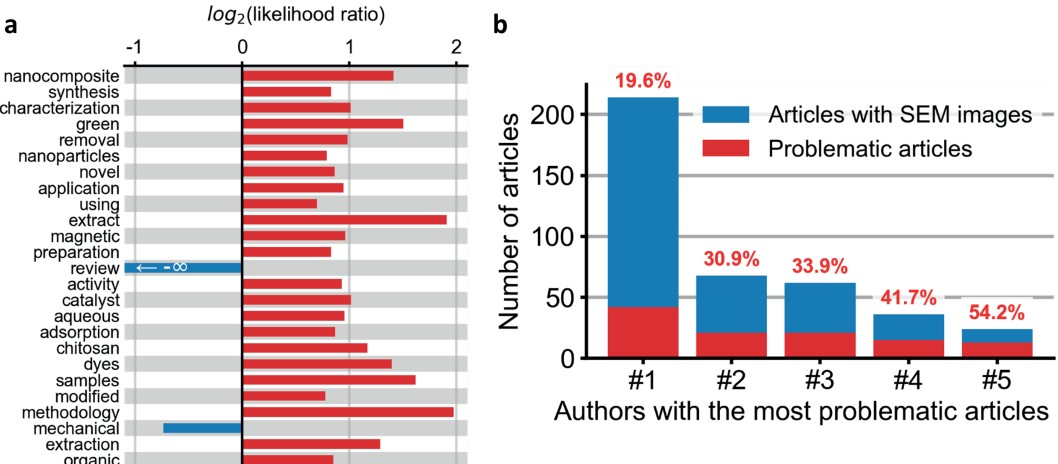

**Fig 7. Systemic characteristics of articles with SEM misidentification. a,** Some terms found in article titles were enriched or depleted among articles that misidentified SEM instruments compared to other articles featuring SEM images. For instance, articles containing 'green' (as in 'green synthesis') in the title were 2.8 times as likely to contain misidentified SEM equipment than articles without 'green' in the title ($p = 1.3 \cdot 10^{-16}$ by two-sided Fisher exact test). We show only non-stopwords found in 100 or more articles with Benjamini-Hochberg false discovery rate < 0.01 by two-sided Fisher exact test. **b,** Rate of misidentification of SEM instrumentation for the five most frequent authors of articles with SEM misidentification.

Author disambiguation problems, especially for Chinese and Korean authors with Romanized names [30], made it difficult to assess how often articles with misidentified SEMs were authored by the same individuals. We were able, however, to identify the most frequent authors of such articles, whose names are unlikely to be shared by multiple individuals. The five most common authors of articles with misidentified SEMs authored between 13 and 42 articles with SEM misidentification. These authors misidentified SEM equipment in 19.6% to 54.2% of their articles containing SEM images (Fig 7b). These authors heavily favored certain journals, publishing up to 16 articles in a single journal in a single calendar year.

Authors of articles with misidentified SEMs were most commonly affiliated to institutions in China (1,027 articles, 42.6%), Iran (668 articles, 27.9%), India (246 articles, 10.3%), Egypt (107 articles, 4.5%) and Saudi Arabia (92 articles, 3.8%) (Fig 8a). 597 articles (25.0%) had authorship from institutions in multiple countries. 497 articles (20.8%) were from institutions in OECD member nations.

Despite their prominence among articles with SEM misidentification, authors from some highly-represented countries actually appear less frequently in the set of papers with SEM misidentification than in the set of all papers with SEM images. For example, we find that even though China is the most frequent country in the set of articles with SEM misidentification, Chinese authors are actually depleted (i.e. under-represented) in this set. The United States, South Korea, and Japan were also under-represented. In contrast, authors from Iran, Egypt, Saudi Arabia, Pakistan and Iraq are over represented among articles with SEM misidentification (Fig 8b).

The SEM images in some of the articles with SEM misidentification show characteristics consistent with coordinated activity by the authors of those papers. For instance, 119 articles contained images marked with "Amirkabir" (referring to Amirkabir University in Tehran, Iran), but only 65 (54.6%) were authored by any individuals affiliated with Amirkabir University. Remarkably, 52 of the 119 (43.7%) articles incorrectly specified the model of microscope

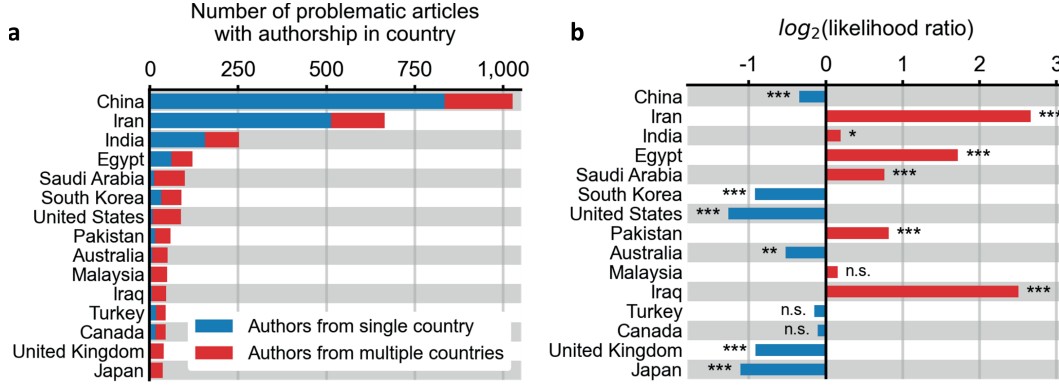

**Fig 8. Systemic characteristics in the affiliations of articles with SEM misidentification. a,** Location of affiliation of authors of articles with SEM misidentification. Many authors are affiliated with institutions located in China, Iran, India and Egypt. A minority of articles had authorship from multiple countries. **b,** Enrichment of location of affiliations of authors of articles with SEM misidentification. Articles with authorship from Iran were 6.3 times as likely to have misidentified their SEM instruments than articles without authorship in Iran (p = 2.9 · $10^{-265}$). n.s. = p > 0.05, * = p < 0.05, ** = p < 0.01 and *** = p < 0.001 by two-sided Fisher exact test.

used, a rate of misidentification nearly double the average for our corpus. This and other cases of image and textual similarity are detailed in S7 Table and S8 Table.

## Limitations

Our analysis is not without limitations. First, the size of image files for figures across the scientific literature increased substantially over the time period surveyed (Fig 2a and S2 Fig). As a result, brandmarks are likely more readily identifiable via OCR in more recent articles than in older ones. Thus, we cannot conclusively determine whether the problem has become worse in recent years. Similarly, image sizes varied considerably across journals and publishers, which may mislead comparative analysis of SEM misidentification rates.

Second, we do not extract any text or figures from articles' supplementary material, where there could be additional SEM images or reporting of SEM equipment. We arrived at the decision to exclude supplementary materials because journals and publisher application programming interfaces (APIs) did not make supplementary material available in a standardized format, unlike the main text and main text figures. Processing information from supplementary materials could recover more instances of SEM misidentification and might reclassify some cases where we found no identification of SEM instruments. However, manual inspection of articles (including inspection of supplementary materials) labeled as having SEM misidentification only yielded additional context in a minority of cases (Fig 5a). Thus, inclusion of supplementary materials is unlikely to alter false discovery rates.

Third, because different journals, editors, reviewers, institutions and authors may hold distinct preferences concerning the inclusion of metadata banners in published articles, some journals and author nationalities may be artificially over- or under-represented among articles with SEM misidentification.

Fourth, SEM misidentification could be much more prevalent than what we report. While our manual evaluation of a subset of articles with potential SEM misidentification demonstrates that our semi-automated pipeline has a low false discovery rate (Fig 5a), our analysis of existing observations of SEM misidentification made on PubPeer (Fig 5c) indicates that our approach has a high false negative rate. Based on this sensitivity, SEM misidentification

could be more than twice as frequent as what we detect with our pipeline. Improvements on this method to improve sensitivity might involve using a lightweight convolutional neural network, vision transformer or other vision model to identify figures containing SEM images with metadata banners, which could then be manually inspected and annotated instead of relying on OCR to discern brandmarks.

## Discussion

We systematically investigated the misidentification of SEM instruments in more than a million articles from across 50 journals published by 4 different publishers. We found that in 24.7% of 11,314 articles for which instrument metadata could be extracted using our approach, the authors did not report the manufacturer and model of the instrument used and that for 21.2% of the articles, the image metadata did not match the SEM manufacturer or model listed in the text of the manuscript. We also find that articles with SEM misidentification are more likely to have existing observations of improprieties made on post-publication peer review site PubPeer than other MSE articles and that a subset of these articles within the subfield of electrochemistry are more likely to incorrectly estimate the optical band gap if the article features SEM misidentification. Although this analysis is correlative in nature, these findings suggests that SEM misidentification may be a tractable signature for flagging problematic MSE articles.

Despite the limitations discussed in the previous section, our study reveals several facts that warrant action from all members of the MSE research community. The presence of repeated authorship (Fig 7b), shared image watermarks (S7 Table) and shared textual artifacts (S8 Table) strongly suggests that mass-production and submission of fabricated manuscripts is responsible for many cases of instrument misidentification. These text and image artifacts could serve as indicators of potential paper mill provenance, alongside other artifacts like tortured phrases [31] and shared images [14]. Indeed, some of the journals considered in our analysis have been advertised online as available venues for guaranteed acceptance (S5 Fig).

While there may be no consensus in the field concerning the inclusion of metadata in images, either in supplementary materials or the main text, it is clear that by failing to request such information one makes it impossible to determine whether experimental instrumentation is misreported. For this reason, availability of metadata has been recognized as essential for ensuring image integrity and adherence to FAIR principles (findability, accessibility, interoperability and reusability) [32–36]. A more straightforward matter may be the establishment of more stringent standards for image resolution in published research (Fig 2 and S2 Fig). Low resolution images make it more difficult to identify reporting inconsistencies and image manipulation. Indeed, low standards for image resolution are likely to challenge recent efforts to automatically screen published and unpublished manuscripts for duplicated images [16,37].

## Supporting information

**S1 Fig. Time series of the number of articles published using SEM annually.** Yearly trends were obtained from each listed literature aggregator with a search of "Scanning electron" in all fields on January 24, 2024.
(TIFF)

**S2 Fig. Extension of Fig 2. a,** Images in which OCR was able to extract brandmarks (red boxes) were initially larger (in terms of image width) than images in which no brandmarks were identified (white boxes), but this difference is negligible in recent years. **b,** Image size (in terms of image height) varied considerably across publishers and within Elsevier's portfolio.

Center line shows median, boxes show inter-quartile range, whiskers show 2.5th percentile and 97.5th percentile.
(TIFF)

**S3 Fig. Extension of Fig 5d.** Pre-existing Pubpeer comments were more frequently found on articles with misidentified SEM instruments than other MSE articles. Errors bars show ±1 standard error of the proportion. n.s. = $p > 0.05$, * = $p < 0.05$, ** = $p < 0.01$ and *** = $p < 0.001$ by two-sided Z-test of proportions.
(TIFF)

**S4 Fig. An example of problematic Tauc plots from a published article [38] that has since been retracted.** First, the Tauc plots shown for the three materials to not seem to resemble their corresponding absorbance spectra shown in the top left plot. Second, all three Tauc plots have a y-axis labeled "$\alpha h\nu$ (eV cm$^{-1}$)", which corresponds neither to an indirect ($(\alpha h\nu)^{\frac{1}{2}}$) nor to a direct ($(\alpha h\nu)^2$) allowed transition. Third, the linear fits in the Tauc plots for MnS (lower left) and rGO/MnS (lower right) are arbitrarily applied to the elbow of the curve and not the linear portion. Finally, in all three Tauc plots, the band gap energy $E_g$ is evaluated at the lower limit of the y-axis instead of at $y = 0$. This final error is the specific error for which we annotated our sample of articles with both SEM images and Tauc plots.
(TIFF)

**S5 Fig. Examples of paper mill advertisements for fast-tracked publication, guaranteed acceptance or pre-written manuscripts in journals in which we also found frequent SEM misidentification.** The names of journals that we surveyed are highlighted in red. Advertisements were found on Facebook, Telegram, WhatsApp and paper mill websites.
(TIFF)

**S1 Table. Characteristics of surveyed journals and collected articles.** When we selected Trends in Analytical Chemistry for analysis, we were unaware that nearly every article was a review article and thus would likely not identify SEM instruments in-text. Our findings confirmed this.
(XLSX)

**S2 Table. Strings used for searching figure captions to identify figures that contain SEM images.**
(XLSX)

**S3 Table. Strings used for matching OCR-extracted text from figures to known SEM models and manufacturers.**
(XLSX)

**S4 Table. Strings used for extracting mentions of SEM equipment in full text of articles.** String matching was case-insensitive. The previous and following 150 characters were extracted into a spreadsheet for manual labeling. All strings reflect popular SEM manufacturers or known misspellings.
(XLSX)

**S5 Table. Metadata and our labels for 11,314 articles with OCR-readable SEM brandmarks.** Strings extracted from full text for labeling are not included to respect the licensing terms and conditions laid out in the text and data mining agreements established by the publishers of the journals we surveyed.
(XLSX)

**S6 Table. Results of screening articles for problematic Tauc plots.** We searched the full text of articles for the string 'tauc', returning 356 articles for which SEM images had brandmarks identifiable by OCR and no detected errors in SEM misidentification (brandmarks_id_correct), 128 articles for which SEM images had brandmarks identifiable by OCR and the SEM instrument was not declared in text (brandmarks_no_id), and 160 articles for which SEM images had brandmarks identifiable by OCR and the SEM instrument was incorrectly declared in text (problematic). We randomly sampled a further 300 articles that had SEM images with no identifiable brandmarks and featured 'tauc' in the full text (sem_no_brandmarks). We manually screened each of these articles for whether they actually featured Tauc plots (has_tauc) and whether at least one of these Tauc plots determines band gap at the wrong x axis (not_at_x_axis).
(XLSX)

**S7 Table. Patterns identified in SEM images which indicate shared provenance.**
(XLSX)

**S8 Table. Unexpected textual patterns we identified in our manual labeling of problematic articles.** These patterns likely indicate re-use of text in Methods sections. Some of these patterns may reflect past licensing agreements between SEM manufacturers or local retailers, although we were not able to find evidence of this being the case.
(XLSX)

## Acknowledgments

We thank Dr. Justin Notestein for his helpful feedback, Dr. Thomas Stoeger for assisting us with accessing OpenAlex data and Helio Tejedor Navarro for technical support.

## Author contributions

**Conceptualization:** Reese A. K. Richardson, Luís A. Nunes Amaral.

**Data curation:** Reese A. K. Richardson, Spencer S. Hong.

**Formal analysis:** Reese A. K. Richardson.

**Funding acquisition:** Luís A. Nunes Amaral.

**Investigation:** Reese A. K. Richardson, Jeonghyun Moon.

**Methodology:** Reese A. K. Richardson, Jeonghyun Moon, Spencer S. Hong, Luís A. Nunes Amaral.

**Project administration:** Luís A. Nunes Amaral.

**Resources:** Luís A. Nunes Amaral.

**Software:** Reese A. K. Richardson, Jeonghyun Moon, Spencer S. Hong.

**Supervision:** Luís A. Nunes Amaral.

**Validation:** Reese A. K. Richardson, Spencer S. Hong.

**Visualization:** Reese A. K. Richardson, Luís A. Nunes Amaral.

**Writing – original draft:** Reese A. K. Richardson, Luís A. Nunes Amaral.

**Writing – review & editing:** Reese A. K. Richardson, Jeonghyun Moon, Spencer S. Hong.

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
