## [Decision Letter · Decision Letter 0]

4 Jul 2025

PONE-D-25-13149Widespread misidentification of SEM instruments in the peer-reviewed materials science and engineering literaturePLOS ONE

Dear Dr. Amaral,

Thank you for submitting your manuscript to PLOS ONE. After careful consideration, we feel that it has merit but does not fully meet PLOS ONE’s publication criteria as it currently stands. Therefore, we invite you to submit a revised version of the manuscript that addresses the points raised during the review process.

We look forward to receiving your revised manuscript.

Kind regards,

Zeheng Wang

Academic Editor

PLOS ONE

Journal Requirements:

We thank Dr. Justin Notestein for his helpful feedback, Dr. Thomas Stoeger for assisting us with accessing OpenAlex data and Helio Tejedor Navarro for technical support. R.A.K.R. was supported in part by the National Institutes of Health Training Grant [T32GM008449] through Northwestern University’s Biotechnology Training Program. R.A.K.R. gratefully acknowledges funding from the Dr. John N. Nicholson fellowship from Northwestern University; Moderna Inc., Identifying bias and improving reproducibility in RNA-seq computational pipelines. J.M. gratefully acknowledges funding from the Weinberg College Baker Program in Undergraduate Research. S.S.H. gratefully acknowledges support from the Ryan Fellowship and the International Institute for Nanotechnology at Northwestern University. L.A.N.A. gratefully acknowledge funding from SCISIPBIO: a data-science approach to evaluating the likelihood of fraud and error in published studies [1956338].

R.A.K.R. was supported in part by the National Institutes of Health Training Grant [T32GM008449] through Northwestern University's Biotechnology Training Program. R.A.K.R. gratefully acknowledges funding from the Dr. John N. Nicholson fellowship from Northwestern University; Moderna Inc., Identifying bias and improving reproducibility in RNA-seq computational pipelines. J.M. gratefully acknowledges funding from the  Weinberg College Baker Program in Undergraduate Research. S.S.H. gratefully acknowledges support from the Ryan Fellowship and the International Institute for Nanotechnology at Northwestern University. L.A.N.A. gratefully acknowledge funding from SCISIPBIO: a data-science approach to evaluating the likelihood of fraud and error in published studies [1956338].

5. Please amend your list of authors on the manuscript to ensure that each author is linked to an affiliation. Authors’ affiliations should reflect the institution where the work was done (if authors moved subsequently, you can also list the new affiliation stating “current affiliation:….” as necessary).

6. Please amend either the abstract on the online submission form (via Edit Submission) or the abstract in the manuscript so that they are identical.

7. Please remove your figures from within your manuscript file, leaving only the individual TIFF/EPS image files, uploaded separately. These will be automatically included in the reviewers’ PDF**.**

Additional Editor Comments :

This manuscript is exceptionally well-written, highly original, and thought-provoking. It attracted an unexpectedly large number of reviewers, all of whom provided very positive evaluations. We have therefore decided to accept the manuscript, pending minor revisions based on the reviewers' suggestions.

Reviewers' comments:

Reviewer's Responses to Questions

**Comments to the Author**

1. Is the manuscript technically sound, and do the data support the conclusions?

Reviewer #1: Yes

Reviewer #2: Yes

Reviewer #3: Yes

Reviewer #4: Yes

Reviewer #5: Yes

Reviewer #6: Yes

2. Has the statistical analysis been performed appropriately and rigorously? 

Reviewer #1: Yes

Reviewer #2: Yes

Reviewer #3: Yes

Reviewer #4: Yes

Reviewer #5: Yes

Reviewer #6: Yes

3. Have the authors made all data underlying the findings in their manuscript fully available?

Reviewer #1: Yes

Reviewer #2: Yes

Reviewer #3: Yes

Reviewer #4: No

Reviewer #5: Yes

Reviewer #6: Yes

4. Is the manuscript presented in an intelligible fashion and written in standard English?

Reviewer #1: Yes

Reviewer #2: Yes

Reviewer #3: Yes

Reviewer #4: Yes

Reviewer #5: Yes

Reviewer #6: Yes

5. Review Comments to the Author

Reviewer #1: The paper explores the reliability of materials science and engineering (MSE) research by analyzing the misidentification of scanning electron microscope (SEM) instruments in published articles. A semi-automated approach was developed to scan figures for SEM metadata and compare it with the instrument identified in the text. The analysis, which covered over a million articles published since 2010, revealed that 21.2% contained mismatched SEM metadata, while 24.7% did not report some of the instruments used. I appreciate the great efforts made by the authors.

Through their analysis, the authors find the potential risks posed to MSE literature due to SEM misidentification, suggesting that these discrepancies may point to broader concerns such as systematic fraud and the presence of paper mills. Furthermore, articles with SEM misidentifications were more frequently associated with prior observations of improprieties recorded on post-publication peer review platforms. The authros also examined trends in authorship and institutional affiliations, noting that certain regions and institutions had a disproportionately high occurrence of SEM misidentification.

Overall, the merits of the paper is sound and the paper is worth to be published. I believe this paper is crucial in raising awareness about the issue of SEM misidentification and its broader implications for research integrity. By bringing this problem to light, I hope to encourage researchers, reviewers, and editors to place greater emphasis on accurately reporting SEM instruments. Beyond just addressing SEM misidentification, I see this issue as a gateway to uncovering larger systemic problems in scientific publishing, such as research misconduct and the operation of paper mills. This work could inspire further investigations into the reliability of published studies and lead to the development of better tools for identifying problematic research. The semi-automated detection method proposed here has the potential to be implemented by journals and publishers, strengthening peer review processes and improving overall research quality.

Reviewer #2: This manuscript presents a systematic investigation into the misidentification of scanning electron microscope (SEM) instruments in the materials science and engineering (MSE) literature. The authors employ a semi-automated pipeline to analyze over a million articles, identifying significant discrepancies between reported SEM metadata in text and images. The study highlights critical issues in scientific reporting and suggests broader implications for research integrity. The work is well-structured, methodologically rigorous, and addresses an important gap in the literature. However, some areas require clarification and further discussion.

Major comments

The high false negative rate (71.1% of PubPeer-identified cases were missed) suggests limitations in the OCR-based approach. The authors should discuss potential improvements, such as alternative OCR tools or manual supplementation, to increase sensitivity.

The reliance on high-resolution images for metadata extraction may bias the results toward recent publications or specific publishers. The authors should further discuss how this bias might affect the generalizability of their findings.

The exclusion of supplementary materials from the analysis is a notable limitation. The authors should justify this decision or address how supplementary data might alter their conclusions.

While the study correlates SEM misidentification with other errors (e.g., Tauc plot mistakes), it does not establish causality. The authors should clarify whether these errors are symptomatic of broader issues (e.g., paper mills) or isolated oversights.

Minor Comments

Some figures (e.g., Figure 2) are dense and could be simplified for better readability. Consider breaking complex figures into subpanels or providing clearer labels.

The term "problematic articles" is used frequently but could be more precisely defined (e.g., "articles with SEM misidentification").

Ensure all references are up-to-date, particularly for tools like Tesseract and OpenAlex, which may have newer versions or related studies.

Overall Recommendation

This manuscript makes a significant contribution to the literature on research integrity in MSE. The methodological approach is robust, and the findings are compelling. However, addressing the limitations and incorporating the suggested improvements would strengthen the manuscript further.

Reviewer #3: The paper by Reese A.K. Anderson, Jeonghyun Moon, Spencer S. Hong, and Luis A. Nunes Amaral, proposed for publication in PLOS ONE, addresses a critical issue in quantitative science: the reproducibility of results. This problem is well-documented in biomedicine and social sciences, where large-scale replication studies have frequently failed to confirm reported findings. However, it has not been extensively studied in the physical-chemical sciences and engineering, despite high-profile retractions in areas such as room-temperature superconductivity, organic electronic devices, and Majorana fermions. With the exponential increase in scientific publications, there is growing interest in developing automated or semi-automated methods to identify erroneous or counterfactual claims in research articles. Publishers are beginning to integrate these methods into their protocols to detect unreliable content in unpublished manuscripts and combat automated mass production of scientific papers.

To address this issue, the authors focus on the use of scanning electron microscopy (SEM) in materials science and engineering (MSE), a widely adopted technique for characterizing materials and their properties. SEM images typically include metadata banners providing details such as the manufacturer, instrument model, and imaging parameters. The study was inspired by reports on the post-publication peer review platform PubPeer, where users observed that metadata from SEM images in many articles did not correspond to the information provided in the text. These discrepancies were often accompanied by other inconsistencies. To investigate this issue, the authors developed a semi-automated method to detect SEM misidentification and applied it to over 1,000,000 articles published in 50 journals from four different publishers, with impact factors ranging from 2.1 to 24.2.

Due to publisher restrictions on large-scale downloads, the study primarily analyzes articles from Elsevier, which has a more flexible data-mining policy. The developed method proved robust against false positives through manual verification, allowing the authors to identify patterns of misattributed SEM brandmarks and models. One example includes systematic errors in the calculation of the optical band gap using the Tauc plot method from electrochemical measurements. The study found that articles including both SEM images and Tauc plot were more likely to miscalculate the band gap if the SEM metadata was misreported. Additionally, recurring trends were identified based on the countries of origin of articles with incorrect SEM identification.

The paper is well-written, with a clear discussion of the methodology and its limitations, and addresses a topic of broad interest to the scientific community. Thus, I consider it suitable for publication in PLOS ONE. However, I would appreciate further clarification on the following points:

1. Selection of initial dataset: The authors state that they begin with an initial dataset of 1,067,108 articles. However, as indicated in Figure 3, only 16% of these contain SEM-related terms in figure captions. This suggests that the majority do not use SEM as a characterization method or include SEM images in the main text or supplementary materials. Merely mentioning "scanning microscopy" does not necessarily imply the presence of SEM images. If the initial dataset is meant to highlight the technique's relevance, this should be made clearer. Otherwise, considering the full dataset skews the perceived percentage of misreporting.

2. Definition of "resolution": The term "resolution" in the article is somewhat ambiguous, particularly in relation to the presence or absence of metadata banners in SEM images. Typically, resolution refers to the number of pixels in height and width. However, the authors focus solely on image height, noting that when it is too low, experimental metadata cannot be recovered. Often, when height is reduced, the banner is intentionally cropped—likely because SEM software does not allow users to adjust font size, making the metadata unreadable in publication formats. If this is the case, further clarification is needed on how cropped images are categorized.

3. Citation trends of misidentified SEM articles: According to the "Systemic Characteristics" section, articles with misidentified SEM metadata receive more citations on average than those with correct information. This is an intriguing observation. Could this be due to the prevalence of open-access journals, where articles with poor experimental details might be more frequently cited? Alternatively, could this indicate another form of misconduct related to citation practices? I would be interested in the authors' perspective on this correlation.

4. Impact factor and SEM reporting: The study finds that lower-impact-factor journals are more likely to contain incorrect or incomplete information, which aligns with expectations. However, I was surprised to see in Table S1 that for Materials Today—the highest-impact journal analyzed—only 2 out of 474 articles containing SEM images included brandmarks, and only one of these was correctly reported. My assumption (and hope) is that the remaining 472 articles included correct SEM information elsewhere, even if it was not retrievable from metadata. The authors discuss the method’s limitations, but in their conclusions, they suggest that these limitations inherently lead to an increased occurrence of misreported SEM data. This statement should be made with more caution.

Overall, the study presents a valuable contribution to the discussion of research integrity and reproducibility in materials science. Addressing the points raised above would further strengthen the clarity and impact of the findings. I also suggest to review the following:

Title. The use of acronyms in the titles should be avoided even if

Figure 1. There is a typo in the text (extant).

Figure 6. In contrast to previous caption, here after any entry a, b, a minuscule letter is used. Revise the uniformity of notation all over the manuscript.

Figure S4. The first phrase of the caption is not clear (no verb).

Reference. The reference style is not uniform across all the entries, and for many of the items the information is not complete (as it happens in mass-produced papers…). A full stop should be included at the end of each item entry.

Reviewer #4: The manuscript presents the differences between the make and model of the SEM mentioned in the text and the data obtained from the image. The study made an investigation of a large number of papers on this subject by a semi-automated approach. Although the results of the study are valuable, I have concerns that some cases require evidence beyond allegations and that these cases may have legal consequences. Therefore, I would like to inform you that the allegations in Figure S5 have the potential to lead to legal proceedings.

-In Figure 4b, the samples labeled as ‘Mfr’, it is not stated but it is understood that the abbreviation of the manufacturer. Also, in the whole text it was mentioned as both ‘make’ or ‘manufacturer’ and the abbreviation is not used. It would be better to use one of them in order to provide the uniformity throughout the text.

-In heading of fig 5. b) ‘For 30 fo the 43’ please correct it.

-In line 240 ‘The five most common authors of articles with misidentified SEMs authored between 13 and 42 problematic articles, respectively.’ What does the respectively correspond?

-In line 289 ‘The presence of repeated authorship (Fig. 7d)’ There is no figure 7d.

- The resolution of the graphs is so low.

Reviewer #5: This is a solid and detailed work about a semi-automatic pipeline to identify SEM instrument information issues in published research papers. The authors focused on the materials science field and examined more than 1 million published works from 50 journals with a diversified set of impact factors. The authors made several significant claims supported by clearly identified numbers. I especially appreciate the: (1) well-prepared figures, (2) well-organized sections, and (3) comprehensive supporting figures and tables.

I support the publication of this work, after some minor revisions are made, including:

1. In Figure 2, image height is used as the indicator of the image resolution. I don’t think this is strictly correct, since usually the image resolution is represented by both height and width pixel counts. I think it would be better if the y axis can be updated to a unit like pixels per inch to indicate the image resolution.

2. Line 162: the claim about the underestimation of global misidentification rate is not necessarily true. The authors should (1) provide an estimation of the misidentification rate of the OCR-unreadable articles and (2) compare it to the OCR-readable misidentification rate. Only then can the authors make an estimated claim of the misidentification rate shift.

3. Paragraph starting from line 193: the authors found the proposed pipeline has a high false negative rate (showed in Figure 5c). How does this observation affect the number of papers with incorrectly identified SEM instruments in Figure 4a? Since this false negative rate is high (71.1%), the authors should consider duplicating some of the discussions from line 278 in this paragraph for easier-accessible information for the reader when reading this part.

4. Other minor issues:

4.1 Figure 2b, include the full form of “OCR” in the legend or in the figure caption.

4.2 In the supplementary spreadsheet, fix the caption for Table S4 (referred as Table S3) and Table S5 (referred as Table 5). The font size can also be consolidated across the sheets.

4.3 Figure 5b caption: “For 30 fo the 43” should be corrected.

Reviewer #6: Very solid piece of work! It raises very important issues about reproducibility and validity of reported results and suggests options to analyse and detect most common faults that prevent FAIR principles.

6. PLOS authors have the option to publish the peer review history of their article (what does this mean?). If published, this will include your full peer review and any attached files.

Reviewer #1: No

Reviewer #2: No

Reviewer #3: No

Reviewer #4: **Yes: **Nazife Nur Yazgan

Reviewer #5: No

Reviewer #6: **Yes: **Aleksandr Mironov

---

## [Author Response · Author response to Decision Letter 1]

3 Jun 2025

See response to reviewer comments in uploaded file

---

## [Editor Report · Decision Letter 1]

5 Jun 2025

Widespread misidentification of scanning electron microscope instruments in the peer-reviewed materials science and engineering literature

PONE-D-25-13149R1

Dear Dr. Amaral,

We’re pleased to inform you that your manuscript has been judged scientifically suitable for publication and will be formally accepted for publication once it meets all outstanding technical requirements.

Kind regards,

Zeheng Wang

Academic Editor

PLOS ONE
---

## [Editor Report · Acceptance letter]

PONE-D-25-13149R1

PLOS ONE

Dear Dr. Amaral,

I'm pleased to inform you that your manuscript has been deemed suitable for publication in PLOS ONE. Congratulations! Your manuscript is now being handed over to our production team.

Kind regards,

on behalf of

Dr. Zeheng Wang

Academic Editor

PLOS ONE